# Modification and Comparison of Methods for Predicting the Moisture Content of Dead Fuel on the Surface of *Quercus mongolica* and *Pinus sylvestris* var. *mongolica* under Rainfall Conditions



**Tongxin Hu** **, Linggan Ma, Yuanting Gao, Jiale Fan and Long Sun ***

Key Laboratory of Sustainable Forest Ecosystem Management-Ministry of Education, College of Forestry, Northeast Forestry University, 26 Hexing Road, Harbin 150040, China; htxhtxapple@sina.com (T.H.); malinggan@nefu.edu.cn (L.M.); 17808052614@163.com (Y.G.); fanjiale@nefu.edu.cn (J.F.)
* Correspondence: sunlong365@126.com

**Abstract:** The surface fine dead fuel moisture content (FFMC) is an important factor in predicting forest fire risk and is influenced by various meteorological factors. Many prediction methods rely on temperature and humidity as factors, resulting in poor model prediction accuracy under rainfall conditions. At the same time, there is an increasing number of methods based on machine learning, but there is still a lack of comparison with traditional models. Therefore, this paper selected the broad-leaved forest tree species *Quercus mongolica* and the coniferous forest species *Pinus sylvestris* var. *mongolica* in Northeast China. Taking surface dead fine fuel as the research object, we used indoor simulated rainfall experiments to explore the impact of rainfall on the surface dead fuel moisture content. The prediction model for surface dead fuel moisture content was modified by the direct estimation method. Finally, using field data, the direct estimation method and convolution neural network (CNN) model were used in the comparison. The rainfall simulation results showed that the indoor fuel moisture content had a logarithmic increasing trend. Rainfall and previous fuel moisture content had a significant impact on the fuel moisture content prediction model, and both the relational model and nonlinear model performed well in predicting fuel moisture content under indoor rainfall conditions. Under field conditions, humidity, temperature and rainfall played a significant role in fuel moisture content. Compared with the unmodified direct estimation method, the modified direct estimation method significantly improved the prediction accuracy and the goodness of fit ($R^2$) increased from 0.85–0.94 to 0.94–0.96. Mean absolute error (MAE) decreased from 9.18–18.33% to 6.86–10.74%, and mean relative error (MRE) decreased from 3.97–17.18% to 3.53–14.48%. The modified direct estimation method has higher prediction accuracy compared with the convolutional neural network model; the $R^2$ value was above 0.90, MAE was below 8.11%, and MRE was below 8.87%. The modified direct estimation method had the best prediction effect among them. This study has a certain reference value for the prediction model of surface fuel moisture content in post-rainfall fire risk assessment and is also of great significance for forest fire management in Northeast China.

**Keywords:** fine fuel moisture content; prediction models; direct estimation method; convolutional neural networks

## 1. Introduction

Fires are occurring around the world, and based on future climate change trends, wildfires will also show an increasing trend [1]. Fuel is one of the basic factors that determine forest fire occurrence and is the material basis of forest fire burning [2]. The moisture content of forest surface fine fuels is one of the important factors affecting the spread and combustion rate of forest fires and is increasingly recognized as the key to comprehensive forest fire management. It has been widely used in fire risk assessment [3]. The moisture

content of fine dead fuels on the surface is sensitive to changes in meteorological factors and the microclimate caused by forest structure [4]. It is an important indicator of forest fire risk level and a key parameter affecting changes in fire behavior [5,6]. Therefore, predicting forest surface dead fine fuel moisture content and improving its precision is of great significance for forest fire prediction.

Rainfall, as a very important meteorological factor, has a significant influence on forest fires. In general, from the perspective of interannual changes, rainfall helps to increase the surface fuel load, thereby increasing the risk of fire. From the perspective of seasonal changes, rainfall during the fire season can increase the fuel moisture content, thereby reducing the fire risk and the harm it brings [7–9]. As rainfall increases, the water absorption capacity of fuel becomes stronger, and the higher the initial moisture content of fuel, the less rain it absorbs [10,11]. The influence of rainfall on surface dead fine fuel moisture content is extremely obvious, but the process is complex and cannot simply equal rainfall directly with the change in moisture content, which also includes processes such as soil absorption and surface evaporation [12]. These complex processes pose certain challenges to predicting the surface fluid moisture content.

Currently, studies on forest fuel moisture content are numerous, and the prediction methods mainly include remote sensing estimation methods, meteorological elements, equilibrium moisture content estimation methods and process models [2,13]. The equilibrium moisture content method is currently the mainstream method for predicting dead-fuel moisture content, and forest fire danger prediction in many countries uses this method to predict fuel moisture content [14]. The Simard method is based on the water loss process of wood to predict the moisture content of combustible materials, and the Nelson method, using a semi-physical model, has high accuracy [15,16]. However, it utilizes temperature and humidity as factors and does not consider rainfall factors, resulting in deviations under rainfall conditions. To predict the fuel moisture content under rainfall conditions, Lopes elaborated on the changes in the relationship between rainfall and fuels [10] and proposed three models for estimating fuel moisture content changes using daily rainfall data. Bilgili, on the basis of the equilibrium moisture content [11], with the combination of wind speed, rainfall, and air pressure difference, built a dynamic model for predicting fuel moisture content. Lee, during the period of spring fire prevention [17], developed a prediction model for the change in fuel moisture content based on different forest densities in the Yongdong region of South Korea using surface dead fuel from the second to sixth day after rainfall as a sample. The model measured the change in fuel moisture content with different diameters.

With the development of computer technology, prediction models such as machine learning have begun to be used to predict the surface fuel moisture content. Among them, neural networks are widely used. For example, Fan used long- and short-term memory network models (LSTM) to predict the fuel moisture content [18] and used random forest, artificial neural networks (ANN) and other methods to conduct research in combination with the data from fuel moisture sticks. The results indicate that LSTM performs well in describing the dynamic changes in time series data. Masinda used the random forest to predict the moisture content of surface fuels in Northeast China [19], and the prediction accuracy was high. Convolutional neural networks, as one of the classic algorithms for deep learning, greatly reduce the number of parameters through convolutional kernels, achieve local correlation, and achieve the effect of parameter sharing compared to traditional neural networks. They can reduce human intervention and maximize the search for information features, thereby enhancing prediction accuracy [20,21]. However, there is currently a lack of research comparing and analyzing machine learning methods such as convolutional neural networks with traditional methods.

Establishing a model of fuel with regard to the changes in the conditions of rainfall is conducive to more accurately understanding the changing situation of fuel moisture content after rain and is more effective in predicting fire danger periods. Promoting the overall accuracy of fuel moisture content and forest fire danger prediction has very important significance [22]. Therefore, this research selects important needle hardwood species of

*Quercus mongolica* and *Pinus sylvestris* var. *mongolica* surface dead fine fuel in Northeast China as the research object, combined with indoor simulation experiments and field data acquisition, and based on the modified direct estimation method, unmodified direct estimation method, and convolutional neural network, the dead fuel moisture content under rainfall conditions is predicted.

## 2. Materials and Methods

### 2.1. Study Area

The research area is located in Maoer Mountain Experimental Forest Farm, Harbin city, Heilongjiang Province (127°34′~127°40′ E, 45°24′~45°33′ N). It is approximately 30 km long from north to south and 26 km from east to west, with a total area of 26,496 hm$^2$. The forest coverage rate is approximately 85%, and the forest stock reaches 2.05 million hm$^2$. The study area contains the flora of Changbai Mountain and has a continental monsoon climate. It mainly has a typical natural secondary forest and plantation forest in Northeast China formed by the destruction of the zonal top-level vegetation broad-leaved Korean pine forest. The tree species are mainly *Quercus mongolica*, *Poplus dividiana* and *Betula platyphylla*. During the 2021 autumn fire prevention season, surface fuels, semihumus, humus and soil were collected in *Quercus mongolica* forest and *Pinus sylvestris* var. *mongolica* forest in Maoer Mountain Experimental Forest Farm.

### 2.2. Experimental Methods

#### 2.2.1. Indoor Simulated Rainfall Experiment

Surface fuels were collected in the experimental stand, and the samples were taken back to the laboratory, placed in an oven, dried at 105 °C for 24 h until they reached absolute dryness, and weighed. Equation (1) was used for calculating the moisture content of fuels:

$$\text{FMC} = \frac{W_H - W_D}{W_D} \times 100\% \tag{1}$$

where FMC is the moisture content of fuel (%); $W_H$ is the wet weight of fuel (g); and $W_D$ is the dry weight of fuel (g).

Different initial fuel moisture content samples were prepared in advance. The impact of rainfall on the dynamic changes in bed fuel moisture content is influenced by the initial bed fuel moisture content. Therefore, different initial bed moisture contents needed to be set for quantitative analysis in the experiment. The condition of the rainfall simulation within the chamber was considered. If dry fuels are selected for testing, the bed moisture content will rapidly increase in a short period of time, which will affect the test results; therefore, the fuel bed was set to four initial moisture content gradients: 5%, 25%, 50%, and 75% [23]. The dried fuels were divided into different weights and placed on the weighing platform. The initial moisture content of different fuels was measured using the formula for calculating their moisture content. The required moisture was added, and the samples were placed in a sealed bag for 24 h to fully absorb the moisture. The sealed bag was used to completely absorb the moisture, and it was also measured before the experiment to determine the fuel moisture content to ensure the accuracy of the fuel moisture content.

Bed compactness refers to the fuel bed volume density and the ratio of grain density. The density of the combustible bed layer was set to three gradients: 0.01, 0.02, and 0.03. Therefore, samples of *Quercus mongolica* with absolute dry masses of 4 g, 8 g, and 12 g were collected, and samples of *Pinus sylvestris* var. *mongolica* with an absolute dry mass of 3 g, 6 g, and 9 g were placed in circular mesh baskets with a diameter of 20 cm and a height of 2.5 cm to achieve different gradient bed fuel compactness. Each gradient was repeated 3 times, and a total of 9 samples were prepared [24].

In the experiment, we needed to set different rainfall intensities. According to the actual rainfall situation in nature, rainfall intensity gradients are divided: light rain, moderate rain, heavy rain and rainstorm. The simulated rainfall experiment was set with treatments

of 2 mm/h, 4 mm/h, 10 mm/h, and 16 mm/h in specific operations to observe the change in fuel moisture content under different rainfall intensities.

We have set up a fuel bed to simulate a non-artificial environment, and placed the samples collected in the field in the order of litter, semi humus, humus, and soil in a rectangular planting basket to simulate the outdoor environment. Then, we loaded the dry weighed test samples into a round mesh basket and inserted them into the rectangular planting basket to simulate actual field measurement conditions. Finally, we simulated rainfall using experimental instruments. We adjusted the appropriate rainfall position, took out the round mesh basket every 10 min, weighed and calculated the moisture content until it no longer changed. The experimental setup is shown in Figure 1.

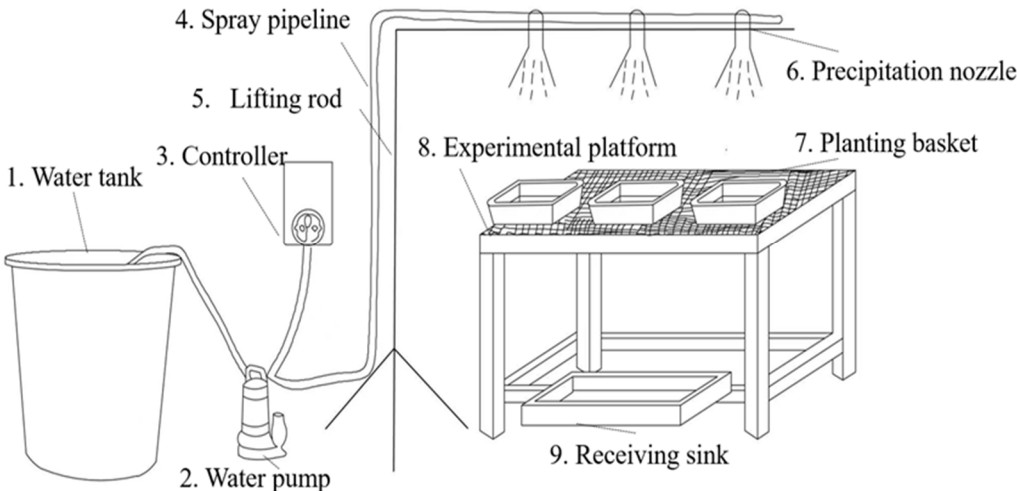

**Figure 1.** Rainfall simulating experiment device.

2.2.2. Field Experiments

During the 2021 autumn fire prevention season, monitoring points were set up at the *Quercus mongolica* forest and *Pinus sylvestris* var. *mongolica* forest in the study area using the random distribution method, and monitoring instruments for surface fuel moisture content and meteorological data were placed [25–28]. The sampling period of this experiment was set to 0.5 h, and monitoring instruments at different slopes were set at the experimental site, which represents the understory environment at different locations. When setting up the instrument, we ensured that the selected points were all located in a standard understory environment. The specific environmental parameters of the sample plot are shown in the table below.

The working principle was to place fuel in a nylon mesh bag, with the lower part in contact with the soil surface, and the upper part without covering, only tightening the bag opening to prevent leaves from entering the mesh bag. The sample was automatically lifted through high-precision tension sensors periodically to achieve continuous measurement of wet weight. Combined with the previously measured dry weight of the sample, real-time moisture content of the sample was obtained [29]. The meters can automatically measure fuel weight, temperature, humidity, wind speed, and solar radiation, and all data can be downloaded with the set application. The monitoring instrument was based on automatic weighing equipment that can continuously monitor the fuel moisture content for a long time and has the function of a small weather station to automatically measure meteorological elements, including air temperature, relative humidity, wind speed, rainfall and solar radiation.

The main working principle of the instrument is to use the weighing method to monitor the moisture content of fuels. First, the fuel is packed in a nylon mesh pocket, the lower part is in contact with the soil surface, with the upper part not covered to ensure its water vapor exchange with the environment. The automated high-precision tensile sensor cyclically lifts the sample and determines the sample weight, achieving continuous

measurement of the wet weight of the fuel sample. With the dry weight of the fuel sample measured separately in the early stage, the real-time sample moisture content is obtained. Meanwhile, meteorological factors such as atmospheric temperature, atmospheric relative humidity, wind speed and light intensity at 1.2 m above the ground are automatically collected [27,28]. The sampling period can be set to 0.05~24 h for any duration, and the step size is set to 0.5 h in this study. To ensure its accuracy, the onsite measurement is compared with the indoor weighing value (accuracy 0.1 g) and parameterized until the error is minimized before the measurement. The study period was from July to October, during which 6000 groups of fuel moisture content data of *Quercus mongolica* and 4500 groups of fuel moisture content data of *Pinus sylvestris* var. *mongolica* were collected. Sample plot information is shown in Table 1.

**Table 1.** Sample plot information.

| Forest Type | Altitude (m) | Location | Aspect | Mean DBH (cm) | Canopy Density | Mean Height (m) | Mean Litter Thickness (cm) |
|---|---|---|---|---|---|---|---|
| *Quercus mongolica* | 395 | Middle | Northeast | 18.40 | 0.70 | 15.20 | 7.20 |
| *Pinus. sylvestris* var. *mongolica* | 380 | Middle | Southwest | 15.80 | 0.40 | 16.47 | 4.90 |

### 2.3. Data Analysis

#### 2.3.1. Constructing a Model for the Relationship between Rainfall and Fuel Moisture Content

Water absorption curves for two types of fuel bed densities under different rainfall intensities were drawn using rainfall experimental data. The influence of different factors on the increase in fuel moisture content in the indoor simulation experiment was carried out by using the ranking of random forest relative importance characteristics [30]. The random forest model performs well in influencing factors ranking and selecting important variables, which is why we used it in this experiment. Selected feature factors improve the model performance.

According to indoor precipitation and growth of fuel moisture content data, the construction model is as follows in Equation (2) [31]:

$$P = t \times R_I \tag{2}$$

where $P$ is rainfall; $t$ is time; and $R_I$ is the intensity of rainfall.

Linear, nonlinear and relational models were carried out according to the simulation data of the relationship between indoor rainfall and fuel moisture content. To prevent overfitting issues during the training process of the selected model in this experiment, the hold-out method was used to directly divide the original dataset into two mutually exclusive datasets. The data was divided into 70% of the training set and 30% of the validation set and several random divisions were used and repeated to take the average value as the evaluation result, in order to obtain a better and more stable model [29].

#### 2.3.2. Wild Fuel Moisture Content Prediction

1.  Direct estimation method

The direct estimation method is a semi-physical method that considers the physical process of fuel moisture diffusion and obtains relevant parameters through experiments [31]. It uses real-time moisture content data and meteorological data to accurately estimate FMC in a short period of time and has good applicability [32]. To make the results more accurate, we selected the Nelson model based on semi-physics and the Simard model based on statistics. These models were used as the equilibrium moisture content response equations in the direct estimation method (hereinafter, they are simply referred to as the Nelson method and Simard method) [15,16]. The parameters of the model were

estimated according to the Least Squares method, and then the prediction model of FFMC was obtained.

We placed the surface fuel at the sampling site in the same mesh bag as was used on the instrument and brought it back indoors to dry at 105 °C for 24 h to obtain the dry weight. Next, we brought the weighed fuel back to its original location and selected some samples around the sample plot, and obtained their real-time fresh weight according to the sampling time interval set by the instrument. Finally, the fuel moisture content was obtained with the same sampling period as the instrument. It was compared with the automatic measurement results of the instrument in the laboratory and used to calibrate the instrument.

This method is mainly based on the differential equation of surface fuel moisture content proposed by Byram, as shown in Equation (3):

$$\frac{dM}{dt} = \frac{E - M}{\tau} \tag{3}$$

where $\frac{dM}{dt}$ represents the change in the moisture content of the fuel in period t; $M$ represents the moisture content of the fuel (%); $E$ represents the equilibrium moisture content of the fuel (%); and $\tau$ indicates the fuel material time lag (h).

The equilibrium moisture content in the above formula is calculated either by the Nelson model or by the Simard model. The Nelson equilibrium moisture content model is shown in Equation (4):

$$E = \alpha + \beta log\left(-\frac{RT}{m}logH\right) \tag{4}$$

where $R$ is the universal gas constant with a value of 8.314 J·K$^{-1}$·mol$^{-1}$; $T$ is the air temperature (K); $H$ is the relative humidity (%); m is the relative molecular mass of $H_2O$, with a value of 18 g·mol$^{-1}$; and $\alpha$ and $\beta$ are parameters to be estimated.

The Simard equilibrium moisture content model is shown in Equation (5):

$$E = \begin{cases} 0.03 + 0.2626H - 0.00104HT & H < 10 \\ 1.76 + 0.1601H - 0.0266T & 10 \leq H \leq 50 \\ 21.06 - 0.4944H + 0.005565H^2 - 0.00063HT & H \geq 50 \end{cases} \tag{5}$$

where $E$ is equilibrium moisture content (%); $T$ is air temperature (°C); and $H$ is relative humidity (%).

Substituting the equilibrium moisture content Equations (4) and (5) into Equation (3) yields the following equation:

$$m_i = \lambda^2 m_{i-1} + \lambda(1-\lambda)E_{i-1} + (1-\lambda)E_i \tag{6}$$

where $m_i$ is the FMC at time $t_i$ (%); $m_{i-1}$ is the FMC at time $t_{i-1}$ (%); $E_i$ is the equilibrium moisture content at time ti (%); $E_{i-1}$ is the equilibrium moisture content at time $t_{i-1}$ (%); $\lambda = \exp(-\delta t/(2\tau))$; $\tau = -\delta t/(2\ln\lambda)$; and the time step in our study is 0.5 h, so $\Delta t$ is 0.5 h.

2.　modified direct estimation method

The prediction model of fuel moisture content after rainfall is:

$$M_i = \lambda^2 m_{i-1} + \lambda(1-\lambda)E_{i-1} + (1-\lambda)E_i + FMCI_p \tag{7}$$

where $FMCI_p$ is the part where the moisture content of fuels increases due to rainfall.

According to the actual moisture content data in the field, the nonlinear estimation method was used to fit the model parameters and obtain their parameters. At the same time, due to the selection of models with different sample parameters, in order to better observe the performance of the model on different samples, the data was divided into a 70% training set and a 30% verification set based on time series to obtain a better and more stable model.

3. Convolutional neural network model (CNN)

The convolutional neural network (CNN) is one of the classical algorithms in the field of deep learning [33,34], which has the characteristics of local connection and weight sharing and can perform high-dimensional mapping processing of raw data and effectively extract data features [35]. The CNN consists of a convolutional layer, a pooling layer, and a fully connected layer. In the convolutional layer, the local connection of neurons and the weight-sharing mode of the convolution kernel can greatly reduce the number of parameters in the training process and improve the training speed of the model. In the pooling layer, through the abstract understanding of the original data, the feature dimension is reduced, which effectively reduces the number of training parameters, reduces the degree of model overfitting, and improves the extraction efficiency of feature data. The alternating use of the convolutional layer and pooling layer can not only maximize the effective extraction of potential features of the input data but also reduce the error caused by artificial extraction of features [20]. In this study, convolutional neural network prediction mainly utilizes the torch package in R. The schematic diagram of the convolutional neural network is shown in Figure 2:

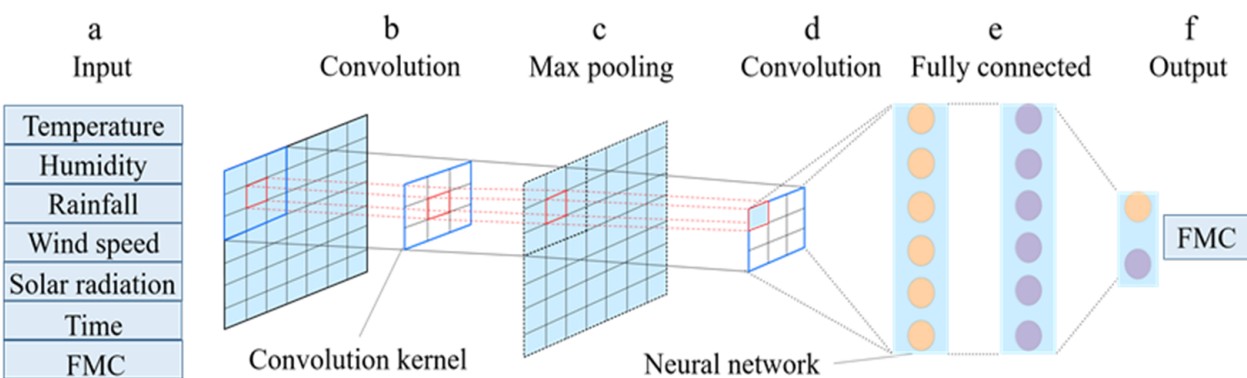

**Figure 2.** Schematic diagram of the convolutional neural network process.

2.3.3. Model Evaluation and Comparison

The goodness of fit $R^2$, the mean absolute error (MAE) and the mean relative error (MRE) of the three models were calculated, and a 1:1 scattered point fitting diagram of the prediction model of fuel moisture content was drawn with the measured value as the abscissa and the predicted value as the ordinate for model evaluation and comparison. Finally, we compared and analyzed the prediction effect through the time series of different model results. The calculation formula is as follows:

$$R^2 = 1 - \frac{\sum_{i=1}^{n}(M_i - M_j)^2}{\sum_{i=1}^{n}(M_i - \overline{M_i})} \tag{8}$$

$$\text{MAE} = \sum_{i=1}^{n}|M_i - M_j| \tag{9}$$

$$\text{MRE} = \sum_{i=1}^{n}\frac{|M_i - M_j|}{M_i}*100\% \tag{10}$$

where $M_i$ indicates the measured value of the moisture content of fuels (%); $M_j$ indicates the predicted value of moisture content of fuels (%); and $\overline{M_i}$ indicates the measured average moisture content of fuel.

### 3. Results

*3.1. Dynamic Changes in Indoor Fuel Moisture Content*

The dynamic changes in surface fuel moisture content of *Quercus mongolica* under four different rainfall intensities, three bed compactness values (0.01, 0.02, 0.03), and four initial moisture contents (5%, 25%, 50%, 75%) are shown in Figure 3. We can find that the *Quercus mongolica* fuel bed increases its moisture content in logarithmic form with the passage of time. After reaching the saturation moisture content, it stops rising and shows a fluctuating growth trend of first fast, then slow. The intensity of rainfall decreases, and the time required to reach the saturation moisture content also increases. To some extent, it is difficult to reach the same saturation moisture content as the intensity of large rainfall. Under different rainfall intensities, the water absorption rate of the bed layer also varies. Generally, when the rainfall intensity is high, the water absorption rate of fuels is also higher in the initial stage. With the increase in rainfall, the water absorption rate curve of the fuel bed gradually flattens out in the later stage. The saturated moisture content of surface fuels in *Quercus mongolica* is generally above 300% after rainfall. The overall trend of moisture content in fuels is the same for different bed densities, with a more rapid increase in moisture content at a bed density of 0.02. Under different initial moisture content treatments, the water absorption rate of the fuel bed with an initial moisture content of 75% is significantly lower than that of the fuel bed with an initial moisture content of 5%.

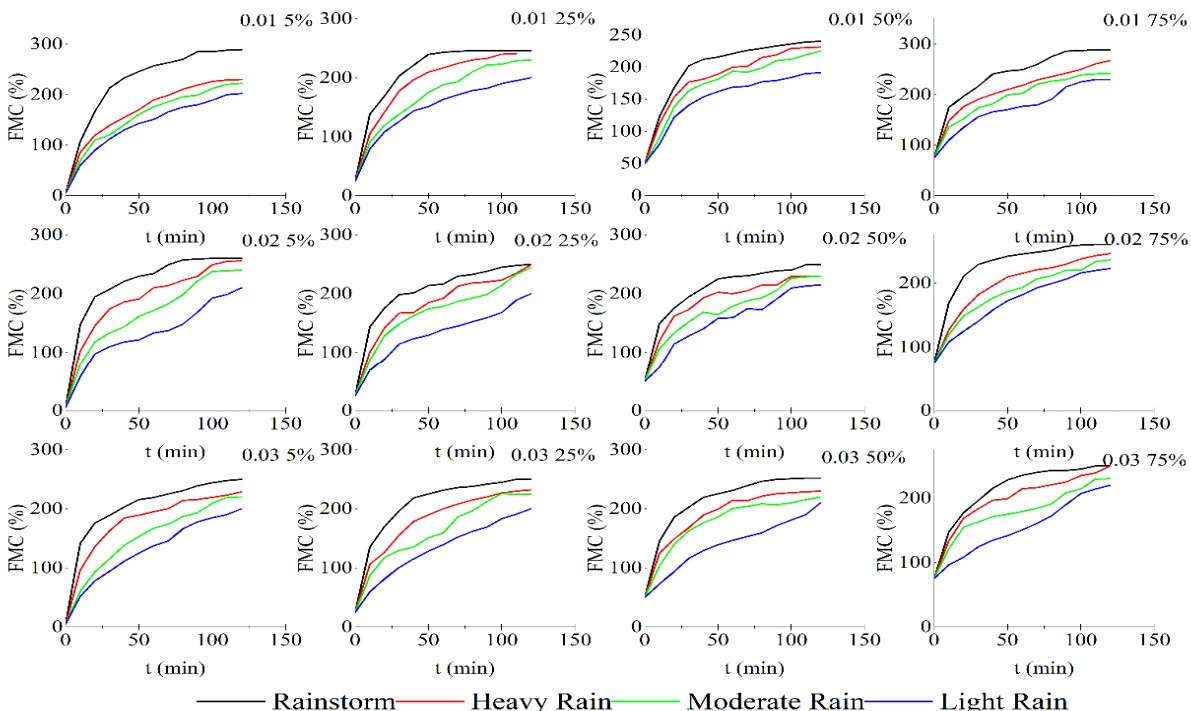

**Figure 3.** Dynamic variation in moisture content in the fuel bed of *Quercus mongolica* under different rainfall intensities.

The dynamic changes in surface fuel moisture content of *Pinus sylvestris* var. *mongolica* under four different rainfall intensities, three bed compactness values (0.01, 0.02, 0.03), and four initial moisture contents (5%, 25%, 50%, 75%) are shown in Figure 4. It shows that the trend of the fuel bed of *Pinus sylvestris* var. *mongolica* is similar to that of *Quercus mongolica* under different treatments, and the trend of the *Pinus sylvestris* var. *mongolica* fuel bed also shows logarithmic growth with increasing time. The saturated moisture content of the *Pinus sylvestris* var. *mongolica* bed under rainfall conditions is more than 150%, and its saturated moisture content is far lower than that of the *Quercus mongolica*. The treatment of different initial moisture contents of fuels shows that the larger the initial moisture content

is, the smaller the growth space for the moisture content of fuels, the lower the water absorption rate, and the greater the difference between the moisture content curves of fuels.

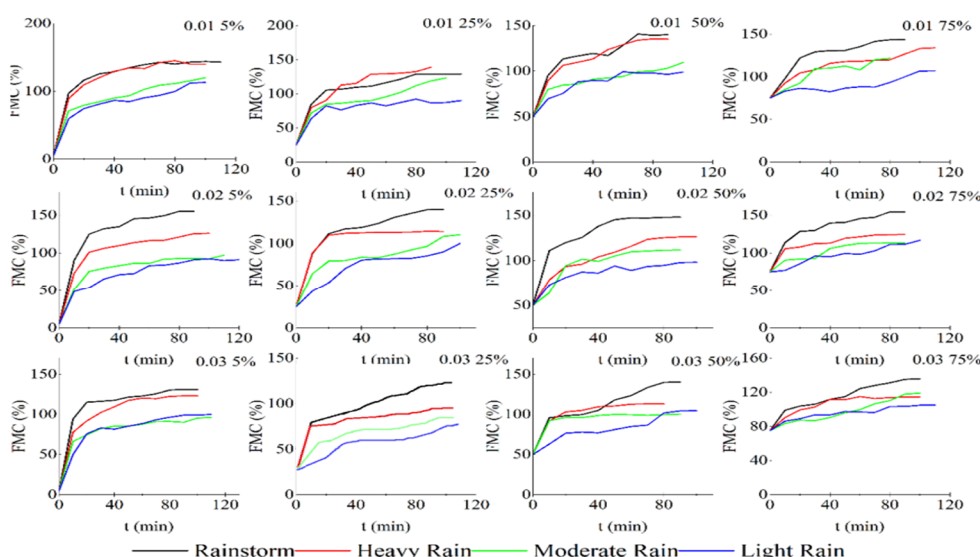

**Figure 4.** Dynamic variation in moisture content in the fuel bed of *Pinus sylvestris* var. *mongolica* under different rainfall intensities.

### 3.2. Ranking of Factors Influencing the Growth of Moisture Content of Fuels

Figure 5 shows that according to the relative characteristic importance of random forests, in the indoor simulated rainfall experiment, under the same rainfall conditions, the previous fuel moisture content is the primary factor affecting the growth of the moisture content of the fuel bed, while the initial moisture content and compactness are not very important to the growth of the fuel moisture content. Compared with the *Quercus mongolica* fuel bed, the results are similar in the fuel bed of *Pinus sylvestris* var. *mongolica*, and the previous fuel moisture content also has an important impact on the increase in the moisture content of the fuel bed of *Pinus sylvestris* var. *mongolica* under rainfall conditions.

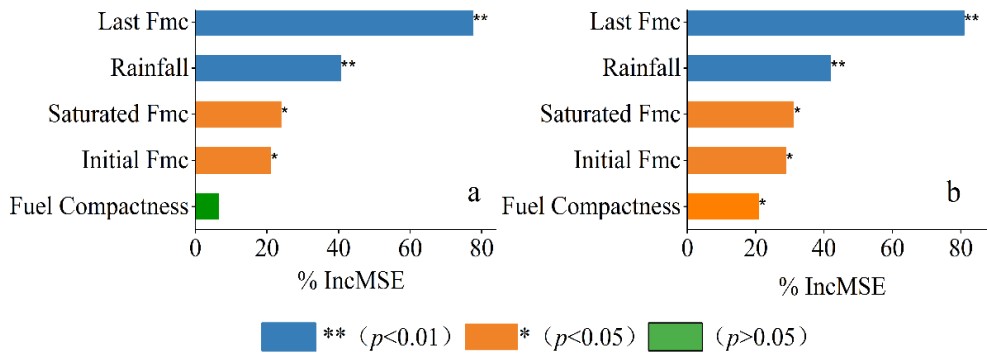

**Figure 5.** Ranking of importance of the drivers of moisture content growth in different stands in indoor simulation experiments ((**a**): *Quercus mongolica*; (**b**): *Pinus sylvestris* var. *mongolica*; and Fmc: fuel moisture content).

### 3.3. Rainfall–Fuel Moisture Content Growth Model

A linear model, a nonlinear model, and relationship model were established for two types of fuels according to the important factor data obtained in the rainfall experiment on the influence of the growth of moisture content of fuels. The analysis shows that in the prediction results of the moisture content of *Quercus mongolica*, the relational model performs better, while in *Pinus sylvestris* var. *mongolica,* the nonlinear model performs better,

and the overall model of *Pinus sylvestris* var. *mongolica* performs better than that of *Quercus mongolica*, and the specific equation is shown in Table 2.

**Table 2.** Model of moisture content growth of fuel due to rainfall.

| Fuel Type | Model | Equation | $R^2$ | MAE (%) | MRE (%) |
|---|---|---|---|---|---|
| *Quercus mongolica* | Linear model | $FMCI_p = 0.408 - 0.43m_{i-1} + 0.021P_i + 0.112smc$ | 0.74 | 32.12 | 24.39 |
| | Nonlinear model | $FMCI_p = 0.987 - 0.92m_{i-1} + 0.209m_{i-1}^2 + 0.01P_i$ | 0.78 | 29.82 | 18.91 |
| | Relational model | $FMCI_p = 0.045P_i(smc - m_{i-1})$ | 0.80 | 11.32 | 9.79 |
| *Pinus sylvestris* var. *mongolica* | Linear model | $FMCI_p = 0.219 - 0.498m_{i-1} + 0.015P_i + 0.002smc$ | 0.82 | 19.61 | 16.44 |
| | Nonlinear model | $FMCI_p = 0.602 - 0.998m_{i-1} + 0.341m_{i-1}^2 + 0.012P_i$ | 0.87 | 12.32 | 18.83 |
| | Relational model | $FMCI_p = 0.034P_i(smc - m_{i-1})$ | 0.73 | 27.54 | 20.03 |

### 3.4. Dynamics of Wild Fuel Moisture Content

During the monitoring period, the moisture content of the surface fuels of *Quercus mongolica* and *Pinus sylvestris* var. *mongolica* showed significant changes over time ($p < 0.05$). The surface fuel moisture content of *Quercus mongolica* fluctuated significantly, showing an overall trend of increasing, decreasing, increasing, and then decreasing (Figure 6). This was because July was rainy and the frequency of rainfall was also high; therefore, the surface moisture content of *Quercus mongolica* was maintained at a high level for several consecutive days. The highest surface moisture content of *Quercus mongolica* was 378.11%, the minimum value was 5.14%, and the average value was 72.73 ± 49.54%. However, in mid-to-late July, without rainfall, the moisture content of fuels remained at a relatively low level, and the change in moisture content was relatively small. In the monitoring data of the moisture content of surface fuel of *Pinus sylvestris* var. *mongolica*, it can be seen that, compared to the surface fuel moisture content of *Quercus mongolica*, its moisture fuel content level was significantly lower during and before rainfall than that of *Quercus mongolica* ($p < 0.05$); the highest moisture content of *Pinus sylvestris* var. *mongolica* fuels was 261.61%, the lowest value was 1.18%, and the average value was 48.16 ± 43.72%. During rainfall, the increase in surface fuel moisture content of *Pinus sylvestris* var. *mongolica* was smaller than that of *Quercus mongolica*, which maintained a uniform level of fuel moisture content of *Pinus sylvestris* var. *mongolica*.

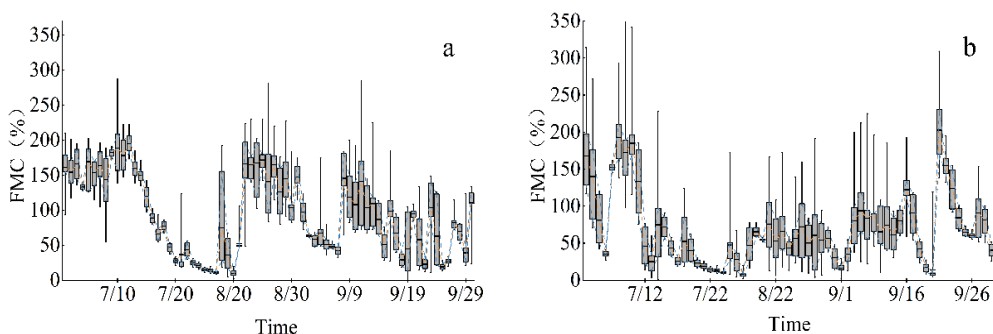

**Figure 6.** Dynamic change in the moisture content of fuel in the wild ((**a**): *Quercus mongolica*; and (**b**): *Pinus sylvestris* var. *mongolica*).

### 3.5. Modified Direct Estimation Method

Based on the prediction model obtained from indoor experiments, the direct estimation method was modified to obtain parameters for different fuel moisture content prediction models. The parameters and evaluation indicators in Table 3 show that, compared to the unmodified model, both modified direct estimation methods improved the degree of fitting in predicting the surface fuel moisture content. For *Quercus mongolica* surface fuels, the fitting degree $R^2$ value of the unmodified Nelson method was 0.85, the fitting degree $R^2$ value of the unmodified Simard method was 0.90, the fitting degree $R^2$ value of the modified Nelson method increased to 0.96, and the fitting degree $R^2$ value of the modified

Simard method increased to 0.94. For the surface fuels of *Pinus sylvestris* var. *mongolica*, the unmodified Nelson prediction fitting degree $R^2$ value was 0.90, the unmodified Simard method $R^2$ value was 0.94, the modified Nelson method prediction fitting degree $R^2$ value increased to 0.95, and the modified Simard method prediction fitting degree $R^2$ value increased to 0.96. It can be concluded that the correction of Nelson's method was more effective in improving accuracy.

**Table 3.** Modified and unmodified direct estimation method model estimation parameters and errors.

| Model | Parameters/Errors | Modified Direct Estimation Method | | Unmodified Direct Estimation Method | |
|---|---|---|---|---|---|
| | | *Quercus mongolica* | *Pinus sylvestris* var. *mongolica* | *Quercus mongolica* | *Pinus sylvestris* var. *mongolica* |
| Nelson | $\alpha$ | 1.081 | 2.144 | 0.595 | 0.445 |
| | $\beta$ | −0.294 | −0.209 | −0.161 | −0.091 |
| | $\lambda$ | 0.982 | 0.994 | 0.899 | 0.921 |
| | $R^2$ | 0.96 | 0.95 | 0.85 | 0.90 |
| | MAE (%) | 8.27 | 6.86 | 18.33 | 9.85 |
| | MRE (%) | 7.84 | 6.12 | 10.91 | 17.18 |
| Simard | $\lambda$ | 0.999 | 0.988 | 0.989 | 0.995 |
| | $R^2$ | 0.94 | 0.96 | 0.90 | 0.94 |
| | MAE (%) | 10.74 | 7.19 | 10.74 | 12.19 |
| | MAE (%) | 14.48 | 3.53 | 14.48 | 3.97 |

*3.6. Convolutional Neural Network Model*

3.6.1. Relative Importance Screening of Meteorological Factors in Random Forest Models

Figure 7 shows that the relative importance ranking of random forest meteorological factors is based on the monitored meteorological data and the surface fuel moisture content data, and the results show that humidity, temperature and rainfall have significant effects on the surface fuel content of *Quercus mongolica* and *Pinus sylvestris* var. *mongolica*, among which humidity is the most significant for the surface fuel content of both. The difference is that in *Quercus mongolica* forests, humidity and rainfall have a greater influence on the moisture content of fuels, while in the surface fuels of *Pinus sylvestris* var. *mongolica*, humidity and temperature have the greatest influence on the fuel moisture content.

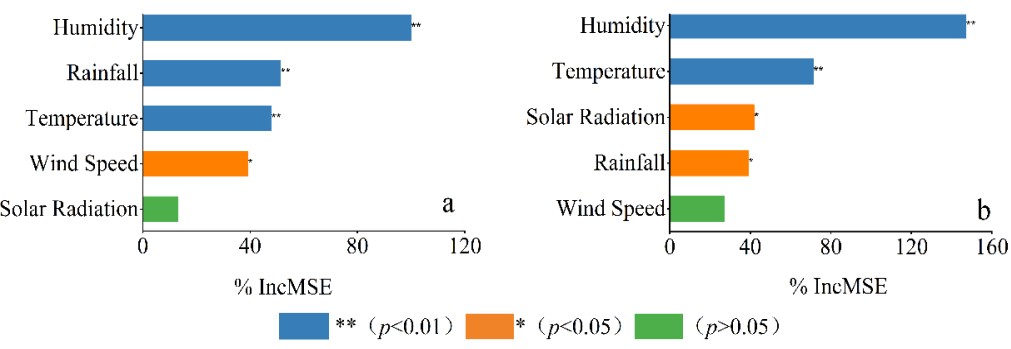

**Figure 7.** Ranking of the relative importance of meteorological variables on the moisture content of fuel on the surface ((**a**): *Quercus mongolica*; and (**b**): *Pinus sylvestris* var. *mongolica*).

3.6.2. Convolutional Neural Network Model Tuning Parameters

In the parameter tuning optimization of the model, it is generally judged whether it is optimal by the function loss between the training set and test dataset. The model tuning analysis for predicting the moisture content of surface fuels is based on convolutional neural networks. The model training loss and testing loss of the moisture content of surface fuels in *Quercus mongolica* had certain fluctuations in the initial stage. But with increasing training times (epochs), the loss function of the training dataset and test dataset

began to decrease gradually, and at the same time, the two curves gradually approached 50 times, indicating that the mean square error was gradually reduced and the model reached a certain effect (Figure 8). For the model parameter adjustment process of using convolutional neural networks to predict the surface moisture content of *Pinus sylvestris* var. *mongolica*, it was similar to the process of *Quercus mongolica* surface fuel moisture content. At the beginning of the iteration, the testing loss was small, but there was a slight difference between the training loss and the testing loss. After continuous iterations, the two gradually approached each other, indicating that its effect was good.

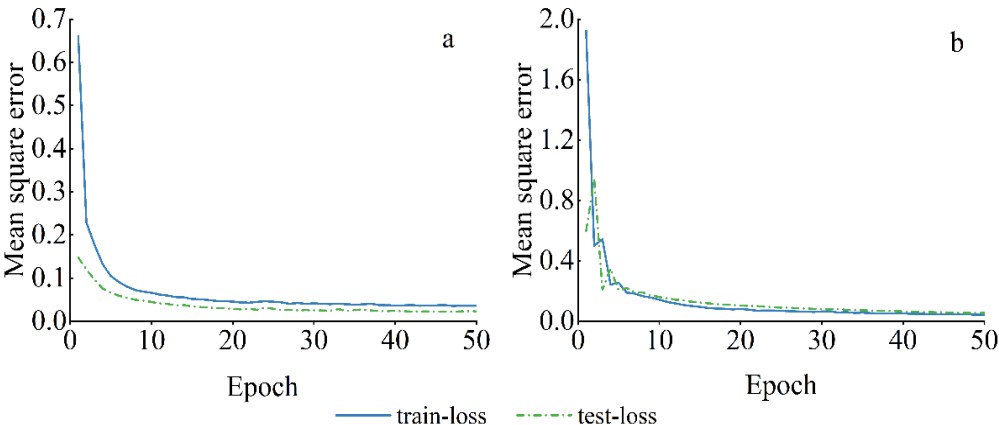

**Figure 8.** Loss results of the training and test sets of different surface fuel convolutional neural network models ((**a**): *Quercus mongolica*; and (**b**): *Pinus sylvestris* var. *mongolica*).

### 3.6.3. Convolutional Neural Network Model Prediction Results

The convolutional neural network model was used to predict the results of the surface fuel moisture content. The surface fuel prediction fitting degree of the *Quercus mongolica* $R^2$ value was 0.93, MAE was 6.05%, and MRE was 8.87%. The *Pinus sylvestris* var. *mongolica* surface fuel prediction fitting degree of the $R^2$ value was 0.90, MAE was 8.11%, and MRE was 4.23% (Table 4). These results indicate that the prediction results of the moisture content of *Quercus mongolica* surface fuels were better than those of *Pinus sylvestris* var. *mongolica* surface fuels.

**Table 4.** Errors of convolutional neural network models for moisture content of fuel on different surfaces.

| Forest Type | $R^2$ | MAE (%) | MRE (%) |
|---|---|---|---|
| *Quercus mongolica* | 0.93 | 6.05 | 8.87 |
| *Pinus sylvestris* var. *mongolica* | 0.90 | 8.11 | 4.23 |

### 3.7. Model Performance Evaluation

### 3.7.1. Comparison of Scatter Fitting for Prediction Results of Different Models

The improvement of the prediction effect is more obvious in the *Quercus mongolica* forest (Figure 9), when compared with the unmodified direct estimation method, regardless of whether the Nelson method or Simard's method improved to a certain extent, especially in some areas with higher moisture content. For the prediction of the convolutional neural network model, the distribution on both sides of the fitted line is also close to uniform, and the $R^2$ value is also better in the *Quercus mongolica* forest (Figure 9).

Figure 10 shows the scatter fitting between the predicted and measured results of the surface fuel moisture content of *Pinus sylvestris* var. *mongolica* in different prediction models, which is similar to the predicted results of *Quercus mongolica*. The modified direct estimation method, whether using the Nelson method or the Simard method as embedded models, performs better than the unmodified direct estimation method. Figure 10 shows that the fitted lines of the modified direct estimation method are closer to the 1:1 line, and

the area of the scattered point distribution is smaller, indicating that the model has better performance. Compared with the unmodified direct estimation method, the improvement in prediction performance is more significant in areas with higher moisture content. For the prediction of the convolutional neural network model, the distribution on both sides of the fitted line is also close to uniform, and the $R^2$ value is also better, but it can be seen that the effect of the unmodified direct estimation method is also better than that of the convolutional neural network under the prediction of the surface fuel moisture content of *Pinus sylvestris* var. *mongolica*.

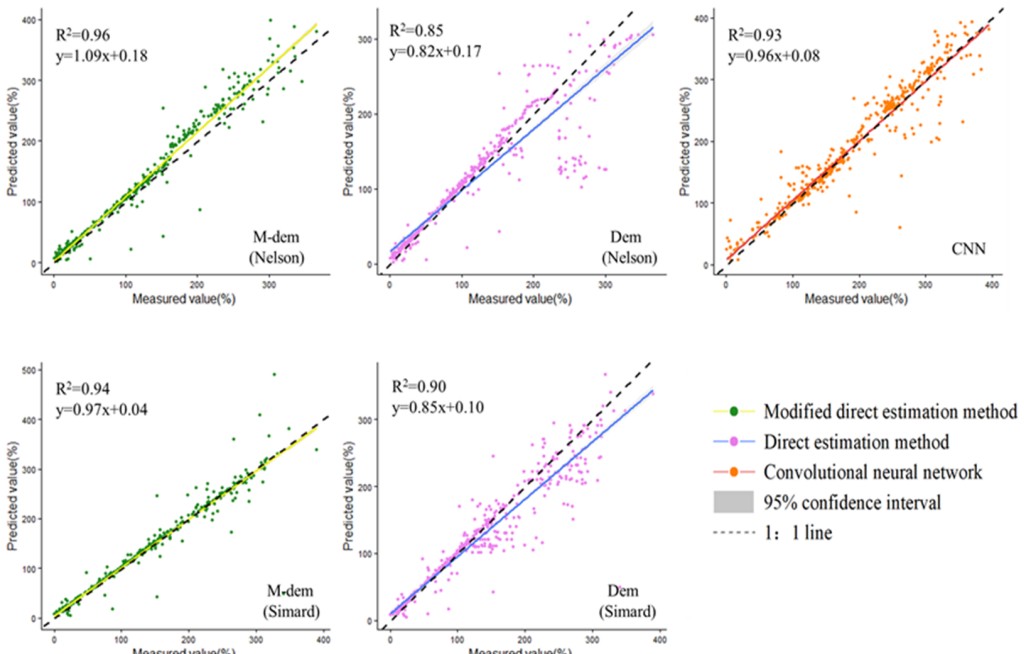

**Figure 9.** Comparison of measured values and predicted values of different prediction models of *Quercus mongolica*.

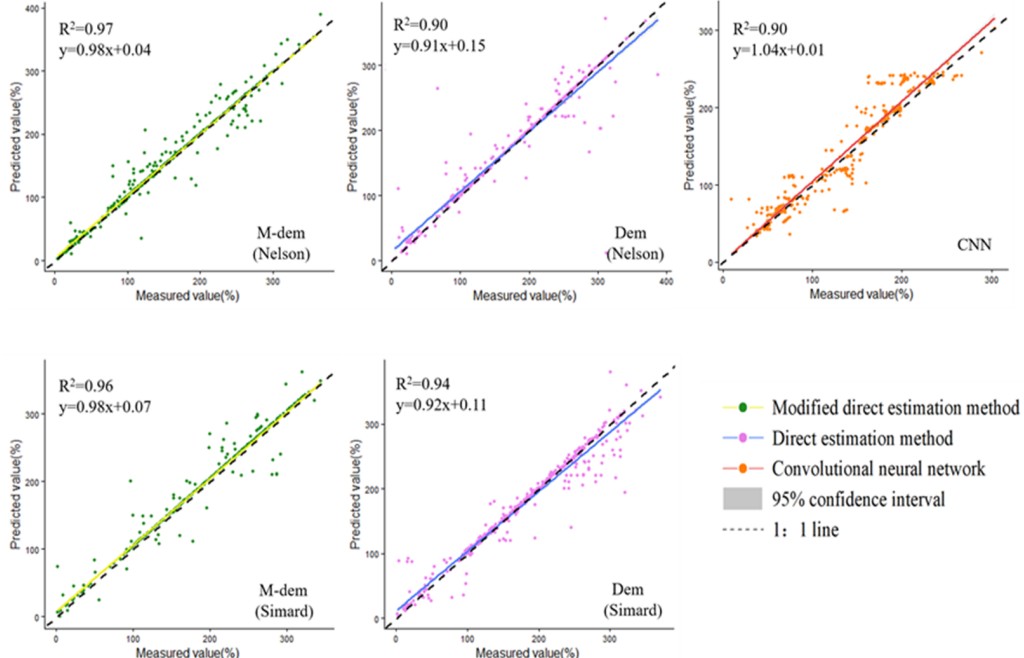

**Figure 10.** Comparison of measured values and scatter of predicted values of different prediction models of *Pinus sylvestris* var. *mongolica*.

3.7.2. Time Series Comparison of Different Model Prediction Results

Figure 11 shows a time series comparison of the results of different models predicting the surface fuel moisture content of *Quercus mongolica*. For different prediction models, the overall trend was consistent on the same date, mainly with significant fluctuations during the rainfall period, while the moisture content level on the majority of dates was low. Due to the different accuracies and characteristics of different models, both models of the modified direct estimation method showed good results (Figure 11). Although the overall trend of the unmodified direct estimation method was consistent, the model's response was slow at the moment when the moisture content curve began to rise (i.e., after rainfall), resulting in a low predicted value at the beginning of rainfall and an overestimation of the predicted value after rainfall ended. For the convolutional neural network, the model had a high overall accuracy, and the curve of the predicted value and measured value was relatively consistent, but it fluctuated in some moments.

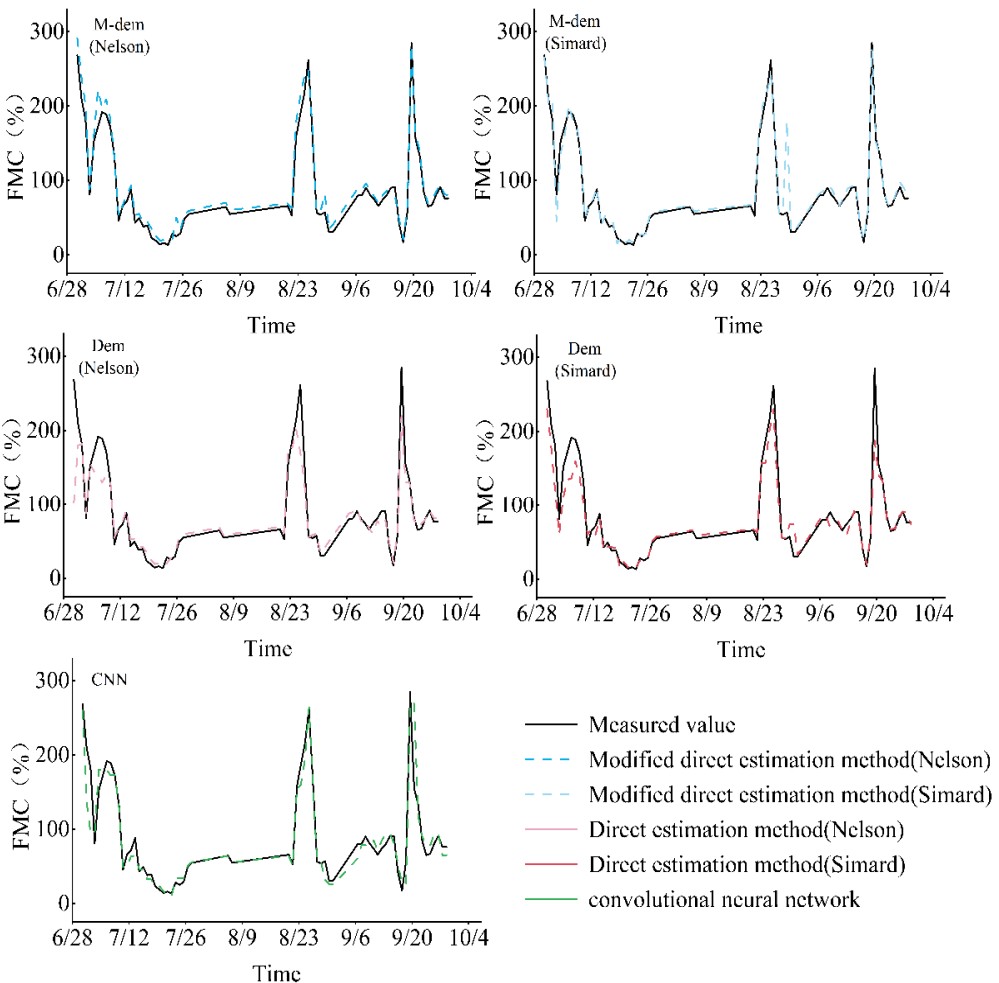

**Figure 11.** Time series comparison of the prediction results of different models of *Quercus mongolica*.

Figure 12 shows a time series comparison of the results of different models predicting the fuel moisture content of *Pinus sylvestris* var. *mongolica*. Several different models had consistent overall performance trends for predicting surface fuels. The two models of the modified direct estimation method showed good results. However, compared to the measured value curve, due to the saturation point of the moisture content of the fuels, the modified direct estimation method may have also overestimated a portion of the moisture content during peak hours, while the unmodified direct estimation method may have also had consistent overall trends. However, after the start of rainfall, the estimated value was low, and after the end of rainfall, there may have been an overestimation phenomenon.

From a comparison of time series curves, the convolutional neural network model was also highly accurate, and it still fluctuated frequently on some dates, but the curve changes were generally consistent.

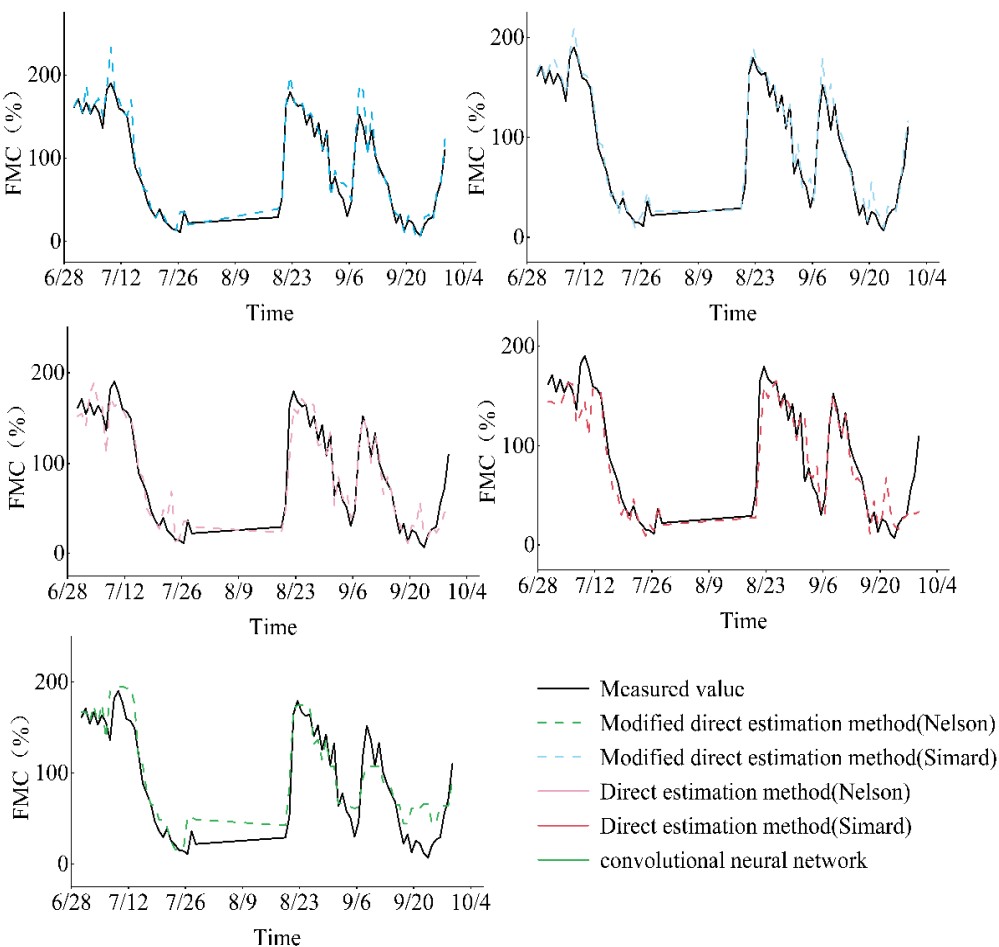

**Figure 12.** Time series comparison chart of the prediction results of different models of *Pinus sylvestris* var. *mongolica*.

## 4. Discussion

### 4.1. The Influence of Rainfall on the Construction of a Model for Predicting Fuel Moisture Content

In the rainfall experiment, the fuel moisture content showed a sharp increase and change in the early stages of the experiment. This was because the surface dead fuel moisture content is composed of water within the leaf structure and surface free water. At the same time, fuels have a large space for the absorption of precipitation under dry conditions, making them prone to sharp peak changes [36,37], which is consistent with the Baksic study on the temporal dynamic changes in fuel moisture content [38]. The difference in moisture content between the two types of surface fuels is mainly due to the larger leaf area and higher decomposition degree of *Quercus mongolica* in terms of the morphology and structure of fuels, resulting in a higher overall moisture content. However, the surface fuels on *Pinus sylvestris* var. *mongolica* exhibit a small needle-like shape and have a low degree of decomposition [39]. At the same time, it can be seen that there is a certain difference between the saturated moisture content of fuels obtained through rainfall experiments and measurements under field rainfall conditions, which may be due to the lag effect of the response of fuel moisture content to precipitation [24].

Our research found that the previous fuel moisture content has a significant impact on the increase in fuel moisture content. This is because under the same rainfall conditions, the higher the previous moment fuel moisture content is, the lower the proportion of rainfall,

and vice versa. This is also fully reflected in many prediction models [40,41]. Rainfall has a significant effect on the increase in moisture content of fuels in experiments, as it can be directly absorbed and stored by the fuel bed, which has been confirmed in many studies [42,43]. The fuel density did not significantly increase the fuel moisture content in this experiment, but in Zhang's study [23], the fuel density had a significant impact on the fuel moisture content. This may be because the fuel density has dual effects, which not only intercept rainfall but also increase the water vapor exchange between fuel beds. In the prediction model of the increase in fuel moisture content on the surface of *Quercus mongolica* under three rainfall conditions, the relational model performs better than the linear and nonlinear models. This may be because the saturated fuel moisture content in the relational model is certain, and the increase in fuel moisture content under rainfall conditions is limited based on the previous fuel moisture content. Therefore, the changes reflected in the relational model between the three are the largest [44]. However, due to the lack of a water dynamics equation that explicitly expresses water absorption and diffusion, it is difficult to improve the accuracy of this model to a higher level [45]. In the growth of fuel moisture content on the surface of *Pinus sylvestris* var. *mongolica,* the nonlinear model performs better. Although the relational model concisely expresses the limited change in fuel moisture content under the condition of rainfall growth, external variations in different fuel beds, temperatures, and humidity can affect the changes in fuel moisture content. Therefore, although the principle of the nonlinear model is simple, its accuracy and flexibility are better than those of the linear regression model, although it is prone to overfitting [46]. Compared with Lopes' linear modeling of the prediction of fuel moisture content under rainfall conditions using three factors—rainfall intensity, rainfall change, and initial combustible moisture content— the $R^2$ value is slightly lower, but the model is more concise and efficient [10]. Therefore, this study aims to further clarify the mathematical relationship between rainfall and fuel moisture content through the establishment of three models, which is conducive to finding representative and inductive physical models of rainfall and fuel moisture content [47].

### 4.2. Driving Factors of Fuel Moisture Content under Rainfall Conditions

In the study of driving factors, humidity is the primary significant factor affecting the fuel moisture content changes in the two forest types, which is consistent with many studies [48,49]. Water molecules in the air constantly move and penetrate surface fuels, causing changes in the fuel moisture content. Rainfall can directly affect the moisture content of fuels, and its impact on the moisture content of *Quercus mongolica* is more important than that of *Pinus sylvestris* var. *mongolica*. This is mainly because of the differences in fuel structure and water absorption and evaporation, which are affected not only by external meteorological factors but also by internal water transfer [50]. The surface fuel bed of broad-leaved forests is relatively loose, and its surface is more susceptible to moisture absorption and solar radiation [51]. In coniferous forests, fuels overlap, their internal water exchange paths are much more complex, and the size and speed of water loss are different [52,53], which is also similar to the results of Bilgili [11]. David used the saturation pressure difference [54], leaf area index, and rainfall to predict the surface fuel moisture content of Brazilian tropical rainforests, resulting in an underestimation of predicted values at higher measured values, which is similar to the research of Masinda [25]. However, there was no underestimation of the impact of rainfall on fuel moisture content in this study, which may be due to the large amount of rainfall period data included in the collection of fuel moisture content and meteorological data, or it may be due to the establishment of a relationship between saturated moisture content and previous fuel moisture content specifically for the increase in fuel moisture content under rainfall in this study. However, at certain times, there may be a slight overestimation phenomenon, which may be due to discontinuous changes in the surface fuel bed structure [55]. Temperature has a significant impact on the moisture content of fuels, and the research results are similar to those previously reported [2,56]. However, it is evident that the increase in fuel moisture content under rainfall conditions

weakens the influence of temperature, but temperature still has a significant impact before and after rainfall [57]. In addition, wind speed and solar radiation have a relatively small impact on the fuel moisture content in the study. This is mainly due to the changes in solar radiation caused by canopy shading in the forest at the same time as the slowing down of wind speed due to vegetation obstruction in the forest [58]. This may also be due to the short time interval for data collection, which cannot reflect the more effective impact of wind on the moisture content of fuel.

*4.3. Model Evaluation and Comparison*

This study constructed three prediction models for surface fuel moisture content for two forest types. Among them, the modified direct estimation method has the highest prediction fit $R^2$ value and low error value, indicating that the model has a good interpretation of the fuel moisture content, low prediction error, and high accuracy. The comparison between the modified direct estimation method and the unmodified direct estimation method also confirms that the modification of the direct estimation method results in better results for the model under rainfall conditions, improving the overall prediction accuracy. The prediction accuracy of *Quercus mongolica* is higher than that of *Pinus sylvestris* var. *mongolica*, which may be due to *Quercus mongolica* inhabiting broad-leaved forests, and the structure of the fuel bed is simpler and more uniform than that of coniferous forests. This result is similar to the research results of Yu [59]. Zhang used the direct estimation method to predict the moisture content of fuels [60], and the error value was much higher than that of this study. This is mainly because the research step size was 24 h, while the step size of this study was half an hour, resulting in higher accuracy. This also confirms that it depends to some extent on the step size for obtaining the moisture content of surface fuels [28,59]. In the direct estimation method, the parameters of the Nelson equilibrium moisture content model need to be estimated based on experimental data, and β is an important parameter in the Nelson model. The value of β can directly reflect the sensitivity of equilibrium moisture content to temperature and humidity. The higher the absolute value of β is, the stronger the sensitivity of fuels to temperature and humidity, and the weaker their water retention ability [16]. In this study, the absolute β value of *Quercus mongolica* was greater than that of *Pinus sylvestris* var. *mongolica*. It also indicated that the water-holding capacity of surface fuels in broad-leaved forests is stronger than that in coniferous forests. This is different from the results of Zhang and Yu [24,59]. This is because different types of fuels and sampling times can affect the physical and chemical properties and structural characteristics of fuels. However, even in the modified direct estimation method, only temperature, humidity, and rainfall were examined, and further exploration is needed for more factor changes and more efficient predictions [61].

Although convolutional neural networks have lower prediction accuracy than direct estimation methods, their fitting degree $R^2$ value is also good, reflecting the advantages of deep learning. Similar to the research results of Lee using machine learning to predict the moisture content of surface fuels with a 10 time lag [62], convolutional neural networks have high accuracy. This is because convolutional neural networks have advantages; they can quickly capture the nonlinear relationship between the fuel moisture content and meteorological factor data and alleviate overfitting problems. Compared with traditional mathematical modeling, it has the advantages of simplicity, rapidity, and accurate modeling and can process a large amount of data, significantly improving the accuracy of the prediction model of the moisture content of fuels [63]. Masinda used random forest to predict the surface fuel moisture content [25], and the $R^2$ value reached 0.86 at the highest value, which was slightly lower than the convolutional neural network, which may be because the convolutional neural network is more suitable for large datasets and has better feature mining ability, but research shows that random forest also has high application potential and development space. The MAE and MRE values in the convolutional neural network model are also low, ranging from 4.23% to 8.87%. However, compared to some machine learning studies, the error is relatively high, which may be related to the selection

of meteorological factors [64], such as the absence of the soil moisture factor in this article. The effect of soil moisture on surface fuel moisture content has been clearly demonstrated, and soil moisture changes are more continuous compared to surface fuels. Therefore, coupling with soil moisture content models can be considered in the future to achieve better results [65,66]. Fan used FSMM-LSTM to predict an $R^2$ value of 0.91 [18], which is comparable in accuracy to the results of this study. This study belongs to the same network structure as convolutional neural networks, but LSTM has certain advantages in timing, while the convolutional neural network proposed in this study was much more concise and faster. At the same time, this study directly collected meteorological factors in the forest microclimate to some extent, avoiding the influence of terrain.

However, in this study, the high accuracy of the direct estimation method and the convolutional neural network model depends on a large amount of data obtained from intensive instrument data collection for model training, and certain outliers in the data reduce the number of samples. Therefore, in future research, the time span should also be increased to obtain a large number of data samples while continuing to explore high-precision models [67]. In this study, it can also be seen that rainfall has a significant impact on the accuracy of model predictions, and its important mechanism should be considered in future research. Furthermore, the impact of seasons and the lag of fuel response to weather conditions on the moisture content of surface fuels should also be considered to improve the accuracy of predicting fuel moisture content.

## 5. Conclusions

This study conducted indoor simulation experiments on the change in moisture content of surface fuels in a *Quercus mongolica* forest and a *Pinus sylvestris* var. *mongolica* forest in northeast China under different rainfall intensities and established models based on the data. The model was obtained by combining indoor simulation experiments, and the direct estimation method model was modified to achieve effective prediction of fuel moisture content in the field. Finally, the dynamic changes and main driving factors of fuel moisture content in the field were analyzed and compared with the unmodified direct estimation method and convolutional neural network. According to the model error and the comparison between measured and predicted values, it was determined that rainfall and the previous fuel moisture content had a significant effect on the fuel moisture content under indoor rainfall conditions. In field experiments, the modified direct estimation method had the best performance, especially in effectively predicting the moisture content of fuels under rainfall conditions. Although the convolutional neural network had slightly lower fitting results in the model, the overall prediction accuracy was relatively high, which can meet the requirements of forest fire risk prediction. The establishment and modification of the model in this study are of great significance for improving the accuracy of the prediction model for surface fuel moisture content under rainfall conditions and have important reference values for forest fire prediction and management in Northeast China.

**Author Contributions:** Conceptualization, L.S. and T.H.; methodology, L.M. and Y.G.; software, T.H. and L.M.; validation, L.S., T.H. and L.M.; formal analysis, L.M. and J.F.; writing—original draft preparation, T.H., L.M. and Y.G.; writing—review and editing, L.S. and T.H.; supervision, T.H.; project administration, L.S. All authors have read and agreed to the published version of the manuscript.

**Funding:** National Key Research and Development Program Strategic International Science and Technology Innovation Cooperation Key Project (2018YFE0207800).

**Data Availability Statement:** The raw data supporting the conclusions of this article will be made available by the authors, without undue reservation.

**Acknowledgments:** We greatly appreciate the "Northern Forest Fire Management Key Laboratory" of the State Forestry and Grassland Bureau and the "National Innovation Alliance of Int J Wildland Fire Prevention and Control Technology", China, for supporting this research.

**Conflicts of Interest:** The authors declare no conflict of interest.

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
