# Peer review of "Modification and Comparison of Methods for Predicting the Moisture Content of Dead Fuel on the Surface of Quercus mongolica and Pinus sylvestris var. mongolica under Rainfall Conditions"

_fire, doi:10.3390/fire6100379_

Round 1
Reviewer 1 Report
Review of “Modification and Comparison of Methods for Predicting the Moisture Content of Dead Fuel on the Surface of Quercus mongolica and Pinus sylvestris var. mongolica under Rainfall Conditions”
This paper explores how fuel moisture content (FMC) models can be improved under rainfall conditions by incorporating indoor simulated rainfall experiments. The authors conducted controlled rainfall experiments on two tree species to analyze moisture absorption dynamics and establish mathematical rainfall-moisture relationships. They leveraged these relationships to modify the direct estimation method for predicting wild fuel moisture, achieving higher accuracy compared to the unmodified approach. The authors also considered using a convolutional neural network for prediction of FMC. They also used a random forest as a means for working out which inputs to the model had the most influence on the predicted FMC.
Overall, the experiments performed yielded an interesting data set and results that showed how current FMC model limitations can be overcome. However, I believe the machine learning methodology could be strengthened to provide more rigorous evidence that these techniques add value over traditional methods like the direct estimation approach. Also, is the data going to be publicly available as a part of this paper, as well as the code?
Main comments:
-
It was not clearly stated if the quantity to be predicted is dead or live fuel. The authors state “fine fuel moisture content prediction” in the abstract, and then “forest surface dead fine fuel moisture content” in line 47. Why not just use FMC throughout and state dead or alive?
-
The authors implemented a convolutional neural network (CNN) to predict fuel moisture content. CNNs are well-suited for extracting spatial features from multidimensional data like images. However, the input data here is temporal meteorological measurements, which does not have inherent spatial relationships. The rationale for choosing the CNN over other neural network architectures was not clearly explained, except that it seems that the authors believe it is more performant than other ML techniques (no benchmarking shown). I think the authors need to consider simpler ML models, for example as was done in these papers, which use random forests, XGBoost, and vanilla feed-forward neural networks. These papers show that forest-based models generally perform as well as or better than hyper-parameter optimized deep learning architectures:
https://iopscience.iop.org/article/10.1088/2632-2153/aba480 and
https://www.mdpi.com/2072-4292/15/13/3372
-
The CNN model configuration like number of layers, filter sizes etc. were not reported (though section 3.6.3 says model tuning parameters but performance metrics are described there instead). So it is unclear if the CNN was optimized for the task or just a basic implementation. Details of the architecture and hyperparameter tuning process should have been provided. Did you use a 1D layer for the time-dependency in your data? If that is the case the authors probably should compare the model to an RNN of their choice to see if the CNN would be an improvement for this kind of time-dependent data.
-
How was the dataset prepared for ingestion into a CNN / RF model? Was there any preprocessing/post-processing of the inputs and predicted FMC values?
-
The authors noted a 70/30 train/validation split, but it was unclear how this was done. It is also not clear from the results whether the splitting was performed randomly or was the time coordinate used? Also the authors did not perform cross-validation to create a model ensemble, so it cannot be determined from the presented results what the uncertainty in model prediction looks like (no error bars are present). No testing data was presented/used either.
-
The random forest model is first mentioned in the Data analysis section. It is unclear to me if it was trained on the same data that the CNN was trained on. Why not compare these two models directly? Both models need to be described in a machine learning section.
-
Feature importance is shown using one method applied to a random forest model, but there is no comparable analysis for the CNN. While the results determined using the RF make sense, it is more common now that multiple, more robust methods, such as permutation importance, SHAP, etc, be investigated together. For example, see reference https://www.mdpi.com/2072-4292/15/13/3372 for a comprehensive analysis.
-
It's unclear if feature selection was performed before RF model training. Reducing the feature space to the most relevant variables could have improved model performance and generalizability.
Other comments:
-
Lines 14-18 in the abstract is run-on sentence that I had to read several times before understanding. Consider rewriting using more but shorter sentences.
-
In line 103, are you referring to live or dead fuel (or both)?
-
Line 196: rephrase the following “relational model modeling”
-
Lines 202-206 are a run-on sentence.
-
Commas are needed in lines 207-210.
-
Line 210: least square → Least Squares
-
Line 225: .. Models (4) and (5) into the discrete Equation (3) yields → equations (4) and (5) into equation (3) yields …
-
Why does the data need to be splitted for fitting the non-linear method?
-
Why does the model schematic show the model predicting two quantities?
-
Line 253: Torch is mentioned as the code but the code is not available.
-
Figure 3: If the figures showed longer than 2 hours, do all the curves plateau, and tend to the same value?
-
The subpanel second from left on the bottom row starts at 50% but it is indicated as being 25% (I think there may be a problem with this subpanel?)
Overall the language was good, however several spots there were run-on sentences (see above). Consider a careful revision as we get closer to publication.
Author Response
Dear editor,
We thank you and two reviewers for the efforts in reviewing and handing our previously submitted manuscript (fire-2543148) entitled “Modification and Comparison of Methods for Predicting the Moisture Content of Dead Fuel on the Surface of Quercus mongolica and Pinus sylvestris var. mongolica under Rainfall Conditions”. After carefully considering all the comments and suggestions from you and reviewers, we have revised the manuscript to address all the comments.
The main changes in the revised manuscript included: 1) clarifying some unclear statements and fine-tuning some paragraphs in the revised manuscript, based on reviewers’ comments; 2) modifying some certain errors in the manuscript; 3) finding a native speaker to modify the manuscript language.
In summary, we were able to address all the major and specific comments. We found the comments to be very helpful and constructive, and they greatly improved our manuscript. In the “Authors’ Response to Editor and Reviewers’ comments”, you can find our detailed, item by item response to each comment.
Thanks once again for your consideration. We looking forward to hearing from you.
Sincerely,
Tongxin Hu, Associate Professor
Faculty of Forestry, Northeast Forestry University, Harbin, Heilongjiang province, China,
Post code-150040
Mobile: +86-451-15046089251
Email: htxhtxapple@sina.com

Reviewer 2 Report
Overall comments:
This manuscript modelled dead fine fuel moisture content under rainfall conditions using multiple models (original semi-physical based models based on the equilibrium moisture, modified semi-physical models as well a machine learning model) for two forest species in Northeast China. As the original semi-physical model didn’t consider the influence of rainfall on the fuel moisture, the authors firstly conducted controlled indoor rainfall experiment to quantify and model the relationship between fuel moisture content and rainfall. Then the best model/relationship from that was used to reflect the increase in fuel moisture content due to rainfall. This relationship was further incorporated into the original semi-physical model for dead fuel moisture content estimation under rainfall conditions. The results were also compared with that from the CNN model against in-situ observations in the study area.
This topic is important as dead fuel moisture content estimation plays an important role in forest fire management. However, this manuscript is not well-written overall. More efforts need to be done before it can be publishable. The presentation and organization of the manuscript are weak, which makes the readers difficult to follow in some parts. I'll provide some of my major comments then move to the specific comments for the improvement.
Major comments:
1. Section 2.2.2
More information is needed in regards to your filed monitoring, for example, the monitoring period, numbers of monitoring sites, site description (fuel bed thickness, fuel structure, canopy cover…) as well as the representativeness of those sites.
You set up the instruments to automatically monitor the weight of fuels in the field to measure dead fuel moisture continuously. It would be good if you add a sketch/figure, which would help readers better understand how you monitor the fuel weight automatically. Besides, more details are needed for your automatic fuel moisture measurement, e.g., the weight of fuels in the mesh pocket, the dimension of the mesh pocket, how would you avoid the leaves falling from the canopy into the pocket since the pocket is not covered… It would good if you would clarify those.
It seems like you collected manual samples for the indoor rainfall experiment. However, I’m wondering whether you also collected manual samples at your monitoring sites and compare that with your automatic measurements from the instruments.
2. Model selection and model assessment
It is not clear in the current version of manuscript in terms of the models you’ve used in your analysis, although you described the models in separate sub-sections under2.3.2. It is hard to follow in the main context when you mention different models as you don’t have a unique name for each model, especially when you mention ‘direct estimation’ vs ‘revised/modified/corrected/uncorrected direct estimation’ throughout the manuscript. I would suggest you revise your method section and present your models in a more efficient way and keep consistency when you refer to these models. In addition, you introduced CNN model in the method but you presented the results from the random forest model, e,g., section 3.6.1. It needs to be justified why you use random forest model for indoor rainfall modelling but CNN model for fuel moisture estimation.
I’m also surprised you didn’t evaluate model performance at the low fuel moisture end (e.g., FMC below 30% or 35%) as low fuel moisture content is more important for fire danger monitoring. As you have the time series data which covers fuel moisture range well, you can potentially evaluate your model performance at low fuel moisture end.
Specific Comments:
L18: Please clarify ‘the direct estimation method’
L21: please clarify ‘last fuel moisture content’
L28-L30: please rephrase this sentence
L53: “With the increase in rainfall, fuel can hold a smaller proportion of rainfall…”, please clarify this
L101: Again, please clarify the ‘modified direct estimation method’ as well as ‘uncorrected direct estimation method’.
L139: lack of unit
L145-L147: Please rephrase this sentence
L150: I found it’s a bit hard to follow this paragraph well in terms of how you deal with rectangular planting basket as well as circular mesh basket? Are you keeping both baskets in your experiment and how did you load your field samples? Please rephrase the whole paragraph.
L185: Please clarify what is ‘DBH’. In addition, more information is need in Table 1, e.g., ‘mean height’ and canopy density.
L285-Figure 3, please clarify 0.01, 0.02 etc, as well as 5%, 25% etc in the figure
L298-Figure 4, same comment as above
L302-Please clarify what is the meaning of ‘last fuel moisture content’
L346-what is the uncorrected model
L373, L390 –- same titles in these two sections
The writing of this manuscript needs to be improved before it can be published.
Author Response

(The authors gave the same response as above.)
